# Benefit of Silver and Gold Nanoparticles in Wound Healing Process after Endometrial Cancer Protocol

**DOI:** 10.3390/biomedicines10030679

**Published:** 2022-03-16

**Authors:** Jakub Toczek, Marcin Sadłocha, Katarzyna Major, Rafał Stojko

**Affiliations:** Department of Gynecology, Obstetrics and Oncological Gynecology, Medical University of Silesia in Katowice, Markiefki 87, 40-211 Katowice, Poland; marcin-sadlocha@wp.pl (M.S.); m.kasia88@gmail.com (K.M.); rafal@czkstojko.pl (R.S.)

**Keywords:** silver, gold, nanoparticles, wound healing, antimicrobial nanomaterials

## Abstract

It is intractable to manage the vast majority of wounds in a classical surgical manner, however if silver, likewise gold and its representative nanoparticles, can lead to the amelioration of the wound healing process after extensive procedures, they should be employed in the current gynecological practice as promptly as possible. Most likely due to its antimicrobial properties, silver is usually applied as an additional component in the wound healing process. In wound management, we obtained various aspects that can lead to impaired wound healing; the crucial aspect for the wound milieu is to prevent the offending agents from occurring. The greatest barrier to healing is represented by the bacterial biofilm, which can occur naturally or in other ways. Biofilm bacteria can produce extracellular polymers, which can then resist concentrated anti-bacterial treatment. The published literature on the use of silver nanoparticles’ utilization in wound healing becomes slightly heterogenous and requires us in difficult moments to set up proper treatment guidelines.

## 1. Introduction

The use of silver for the amelioration of wound healing has been observed since 69 BC. We differentiated various silver characteristics which can be applied in medicine. First of all, silver metal (Ag) has no medical use, while silver ions (Ag^+^) or other forms of silver, as nanoparticles, have a broad antimicrobial spectrum of activity. They are cytotoxic to many bacteria, viruses, fungi, and even yeast which can disturb the healing process [1]. Ionic silver is directly adjacent to DNA and RNA molecules, as well as to various proteins, which eventually leads to apoptotic processes via multiple pathways, for instance, protein and nucleic acid denaturation enhances membrane permeability and has itself a noxious effect on the respiratory chain (electron transport chain). Hence, resistance against silver ions has been sporadically reported. The character of nanoparticle silver (AgNP) creates a unique, enormous potential for antimicrobial effect. In various protocols associated with the breeding of certain bacteria species on petri dish agar plates, for instance, *Staphylococcus aureus*, *Pseudomonas aeruginosa*, and *Escherichia coli*, the zone of inhibition wase indisputably recognized (which means that the nanoparticles have probable properties to inhibit the growth of the bacterial cells) [2,3]. A pivotal aspect of nanoparticles was related to the inhibition of biofilm formation likewise reducing biofilm biomass. As we know, biofilm initiates a beneficial milieu for bacteria to resist antibacterial prevention. Unfortunately, most antibiotic-resistant bacteria are acquired during hospitalization, when a patient becomes colonized by these species of microorganisms after extensive procedures, such as the endometrial cancer protocol. The patient can then require prolonged hospitalization due to wound healing impairment due to the resistant biofilm. The most significant aspect of AgNP is that it can be obtained from yeast culturing, especially the Saccharomyces cerevisiae strain, thus widespread use seems to be easier and it is much cheaper to produce this kind of silver [4,5]. This species of probiotic yeast has silver nitrate reductase properties in the aqueous solution. Production of AgNP were emerge in low and high concentrations of AgNO_3_. In brief, 48 h of yeast culturing at the optimal temperature, at 28 °C degree with shaking, the cell suspension when reaching a certain density becomes centrifugated. The obtained suspension was removed, and the yeast biomass was then resuspended with sterile distilled water and cultured once again for 24–48 h with the shaking process. Overall, the process, as aforementioned, is usually time consuming. If production can progress much faster, then the widespread use of silver and gold products can become easier to distribute in the medical industry for the amelioration of wound healing parameters [6].

## 2. Wound Healing Process after Operation

Skin is the largest organ of the body and simultaneously has various essential functions. In particular it is responsible for sensory, homeostatic, and temperature regulation and acts as a protective barrier against pathogens. Disruption of the integrity of the skin in a particular area determines formation of a wound, which can occur as a part of a certain disease, or be part of an injury or an intentional aim (for instance, an extensive laparotomy incision) [7,8]. A wound can arise from various factors such as the result of surgical intervention, from injury or other factors or conditions, namely diabetes mellitus or vasculopathy. In addition, these patients, especially females, are prone to develop the disease, endometrial cancer. The vast majority of womens are nicotine abusers, the elderly population or have high-stress vulnerability employment. Other factors which contribute to the wound healing process include the nutritional aspect, immunological status, smoking, diabetes, and obesity. Hypertension can also have an important impact on the course of healing [9,10]. Generally speaking, the longevity of the elderly population has increased gradually, which has had an influence on the higher prevalence of non-healing ulcers. In patients, with chronic disabilities, or extensive laparotomic operations, for instance the endometrial cancer protocol, where the uterus is simultaneously dislodged with an adnexal mass, the lymph nodes from the iliac region may be gathered likewise. Skin microcirculation remains crucial for the healing process, and is impacted by impaired vasoregulation, an altered inflammatory response and fewer cells for proper regeneration. The incidence of chronic wounds, likewise wounds following extensive invasive operations, has increased among the older population, which is also associated with greater incidence of cancer and has a high impact on the quality of life [11,12]. Overall, the wound-wide expenditure of the health care system has increased to manage the healing process for ulcers. From year-to-year the costs of the impairment in wound regeneration increases by degrees. This is why the optimal and ideal dressings are still the subject of research programs, to fix the obstacles relating to the wound healing protocol. The exemplary, wound dressing should be compatible with our extracellular matrix and be characterized by biological stability, flexibility, and ability to remove exudate from the wound and to provide a proper, moist milieu for the healing site. The wound should be adequately protected from the external environment and hazards, such as a bacterial infection. The wound itself can be classified as acute (surgical), or chronic (diabetic foot wounds or pressure ulcers). A chronic wound usually is defined as a wound in which restoration of proper skin integrity is unattainable in relation to anatomic and functional integrality within three months. Restoration of tissue integrity and function has been associated with multiple cellular and extracellular pathways through superimposed phases, one step after another, consequently hemostasis follows the inflammatory phase, then the proliferation phase, and finally the remodeling phase. Hemostasis (vascular response) leads to contraction of the blood vessels and the segmental accumulation of the platelets in the fibrin mesh. The proliferative phase then follows, which is an angiogenetic process, and includes the epithelial closure, which eventually leads to wound healing [13,14]. The last phase comprises the remodeling phenomenon, which is achieved by the suitable reorganization, disintegration, and resynthesis of the extracellular matrix which leads to granulation, or scar, tissue. During the wound healing process, multiple factors can participate in the process. Thus, growth factors and cytokines spread at the wound milieu, strictly cooperating in the regulation of the healing site.

Due to this multistep process, various factors can interfere in delaying wound healing, which could result in low cosmetic resolution. From the external wound perspective, appropriate suitable wound healing depends on various parameters, mainly wound size, depth, localization, patient age, and presence of chronic disease. Usually a patient, after the endometrial cancer protocol which comprises removal of the uterus simultaneously with adnexa along with lymph nodes’ specimens harvest, has a chronic disease such as diabetes, vascular disease, neuropathy, immunological deterioration status [15,16]. Mitigation epidermal remodeling along with angiogenetic process induction can decisively lead to the synthesis of new connective tissue. The vast majority of wound dressings have emerged for their protective effects on a wound from infectious hazards rather than to induce renovation process. Recent wound healing approaches encompass the use of autografts, allografts, and cultured epidermal cells. Currently, more than 3000 products have been designed, involving concentrated wound dressings, dressings incorporating growth factors and biological molecules to trigger augmentation of cellular remodeling and ECM synthesis along with skin substitutes merged with patient-derivate [17,18].

## 3. Role of the Silver Products in a Wound Healing Course

Silver in the form of an ion was used in ancient times, especially in Egypt. This ion was utilized particularly to treat difficult wounds, in the form of a wound dressing with the composition of silver ion. Even Hippocrates described, in many of his books, the pivotal effect of silver on the wound healing process. Generally speaking, silver contains many advantageous properties, in particular broad-spectrum antibacterial capabilities, silver-based creams and other ointments, likewise silver nanoparticle products can be exploitable in different remedial forms which can eventually promote the process of wound regeneration [19,20]. Recently, widespread infectious disease, in particular highly resistant microorganisms, has caused pharmaceutic business and scientists to search for the unique antibacterial product. Currently, silver nanoparticles, as well as other biopolymers, have attracted scientific interest. In the last couple of years, a rising number of publications have emerged. Various antibacterial effects of silver have been recognized, decreasing the chance for rising microorganism resistance, as well as enhancing the effectiveness in opposing organisms that are recognized as multidrug-resistant. AgNP properties are becoming really promising, thus if the concentration of an AgNP as a percentage was higher in the cream formula, the wound would demonstrate an augmented deposition of collagen, and the increase in macrophages’ migration and more fibroblasts are transformed to myofibroblasts, for preferential remodeling [21,22]. Multivarious retrospective investigations revealed that the utilization of silver as widespread bandages has been related to reduced inflammation foci and scarring, potentially expelling bacterial outgrowth and promoting the healthy process, enhancing remodeling in the wound area.

Hence, utilization of the nanocomposites immerses within the silver molecules, enhances the healing process through the direct expression of collagen and certain growth factors which leads to re-epithelialization, vasculogenesis (neovascularization) and deposition of collagen fibers. Moreover, silver nanoparticles can induce distinctness of the fibroblast to myofibroblast, which is responsible for contraction of the wound and speeds up the course of healing, and in a similar manner can lead to keratinocytes to be stimulated and proliferate and relocate to the required location [23,24]. Nanoparticles of silver induce migration of keratinocytes from the edge of the wound into the center for better wound healing. Some scientists debate concerning AgNP that nanoparticles amplify the course of wound healing through the suppression of bacterial outgrowth and generate pro-inflammatory cytokines deliverance. You et al., investigated the action of silver nanoparticles on the course of healing; they found that an appropriate number of nanoparticles can participate in the migration of fibroblasts from the edge of the wound, which can then enhance the level of alpha-smooth muscle actin, and the enhanced fibroblast can differentiate to the myofibroblast and contract the wound which promotes the course of faster remodeling. Over the last few years, interest in silver as an antimicrobial agent has increased in a significant manner. In the wound dressing industry, the silver-based issue is already patented and widely commercialized. In comparison to normal dressings, silver products improve efficacy compared to a standard dressing [25,26]. Synthesized biopolymers are consolidated with bioactive antimicrobials, antibacterial, and anti-inflammatory nanoparticles, and show great augmentation of the wound healing potential, especially in the deployment for extensive surgical wounds after a complicated laparotomic operation such as the ovarian or endometrial cancer protocol. Certain scientists reveal that antimicrobial peptide-AgNP composite has been tested for its wound healing properties and has shown an improved remodeling process without any side effects concerning dermal tissues and in addition escalates the interaction between peptide and lipopolysaccharides moieties on bacteria, consequently disclosing a wide spectrum of activity without encouraging bacterial resistance. The disparate antibacterial outcome of silver can be noticeable, decrease the chance of bacteria inducing resistance and boost effectiveness in opposition to multidrug-resistant microorganism. Increasing the percentage of AgNP in the applied dressing reveals a reduced wound area, and augmentation of collagen deposition associated with migration of macrophages and fibroblasts [27,28]. Scientists consider that maintaining an appropriate concentration of nanoparticles immersed in the wound area can lead to the judicious migration of pro-inflammatory cells, that participate in a local response as the main modulators. The crucial pro-inflammatory cytokines in the wound healing process are IL-6, TGF-a, IL-10, IFN-y, which can be searched by using RT-PCR. A different m-RNA amount can be found in the wound milieu that becomes modulated by silver nanoparticles. The authors also manifested a decreased percentage of scar occurrence after usage of silver nanoparticles. The retro-analytic study performed on the usage of AgNP dressings, displayed antibacterial properties against *Pseudomonas aeruginosa*, and also concordance with fibroblasts. Properties of the anti-biofilm capability of nanoparticular silver have potential interference, inhibition, and regulation of EPS products by bacteria.

Microbial concentration in the wound milieu has crucial aspects concerning healing properties and avoiding incremental costs on antibiotic use. Supervising bacterial outgrowth in the surgical wound becomes fundamental for effective wound renovation. Reduction of the bacterial growth with a dressing comprised of nanoparticles, hastens the wound healing course. Most in vivo investigations reveal proper wound healing features due to intrinsic antibacterial, anti-inflammatory, and hemostatic characteristics. Furthermore, except for antimicrobial properties, silver surgical textiles demonstrates a boost in healing features, as a result, silver exploitation has a positive effect on cell migration and proliferation quality [29,30,31].

## 4. Introduction of Silver Nanoparticles as Promising Antimicrobial Agent

The exploitation of silver and other ions or ingredients was abandoned when antibiotics emerged but, the recent momentous resurgence has been associated with silver use against emergency antibiotic-resistant strains, and has a low tendency to develop resistance. Due to its intrinsic healing potential, also its multisite participation, AgNP exposes a broad-spectrum antibacterial ability in opposition to many microorganisms and reveals the enormous potential to avoid the emerging issue in the microbial resistance case. Nowadays, silver nanoparticles are in increased demand for medical applications, concerning wound dressings after various extensive procedures, artificial implantation, and even utilization in the surgical theatre [32,33]. Further instances include use of AgNP for the implantation of medical devices, to overcome infection and induce the wound renovation process. Moreover, with the tremendous potential of AgNP in biomedical participation, much effort has been put into comprehending the intricate character of their biological activity, which usually depends on various parameters of silver, such as concentration, colloid state, surface coating, dimension, and shape. It is established that nanoparticles of AgNP which are of a smaller diameter have preferable antimicrobial effect in comparison to the large one, Table 1 [34,35]. Beneath 10 nm, the properties of nanoparticles are mostly assigned to the nanoparticles per se, when in large particles, it preponderantly materializes as a silver ion. The mechanism of action with regard to silver nanoparticles in various body systems remains still inexplicable, but it most likely has been assessed that due to the nanometric character and increased surface area, nanoparticles of silver can shatter the membrane, cross the body of the microorganism and complete intracellular annihilation. Total dissociation of LPS and cell removal occurs through membrane protrusion attached to NP, which gains access to the cell by electrostatic alteration. Inducing oxidative stress, metal ion release or other non-oxidative contributions are also means for explaining the antimicrobial mechanism of AgNP [36,37].

The preponderance of AgNP on the bacterial surface leads to the accumulation of silver Ag+ species, which corresponds with the inactivation of the biological machine, such as deoxyribonucleic acid, peptides, and cofactors which are targeted by the AgNP. Silver-binding proteins have the capability to interfere with many necessary processes that maintain bacterial lifespan. In the vast majority of cases, silver disrupts the key enzymes that participate in glycolysis and the TCA cycle and in the oxidative defense system. The biological activity of Ag+ differs according to the specificity of the cell wall in the bacteria, Gram positive or Gram negative. Generally, the most pivotal role of silver in disrupting bacterial growth is the antibiofilm mechanism, likewise the capacity to interrupt the synthesis of exopolysaccharides [38,39]. Another potential mechanism through which silver nanoparticles ions reduce the formation of biofilm is the disruption of Quorum Sensing (QS), especially bacterial gene expression mechanism controlled by small molecules. Ravindran et al., reveal the effect of photo-sensing nanoparticles of silver as anti-quorum sensing, also the antibiofilm agents are opposed to multi-drug resistant pathogens. In comparison to antibacterial which can decrease the speed of outgrowth aspect, silver ions act more non-specifically in many various targets, hence stirring batteries in many components of the metabolism and structures [40,41].

## 5. Gold Nanoparticles Use as Antimicrobial Synthetic Agent

Thus, concerning this review, we will now focus on the wound healing capacity of gold metal nanoparticles, infection prevention being a crucial step in this process. The utilization of some metals as agents which act as antimicrobials can be dated back to ancient civilizations. The last decade has seen an increase in the use of metals and metal nanoparticles to conquer infection. Hence, many metal nanoparticles are now widely available commercially and used more frequently in hospitals for wound dressings [42,43].

Some metal ions have an extensive range of antimicrobial spectrum with multiple cellular targets. This spectrum of action is usually due to the metal’s unique features which permit the interactions with relevant functional groups in biological molecules. The selectivity and specificity of metal attachment, which is strictly associated with its antimicrobial effectiveness. Depending on the properties of the metal, various metals will target alternative functional groups in metabolites, proteins, nucleic acids, lipids, and carbohydrates, as any of these components can cause detrimental cascading reactions in the cell. We can divide the effect of the previous cascade into the direct and indirect membrane impairment, changed membrane potential, and transport across the membrane, protein misfunction and denaturation stirring metabolism, electron transport chain impairment, DNA damage and conformation alteration, inhibition of DNA replication and repair, and carbohydrate degradation [44,45].

The creation or stimulation of reactive oxygen species (ROS) is the subsequent mechanism by which metals interfere with noxious effects. Certain metals are redox-active, with the capacity to donate or accept electrons from different atoms. Redox-active metals lead to reactive oxygen species by catalyzing Fenton reactions. Inhibition of proteins or other compounds related to redox equilibrium or of the electron transport chain may also indirectly lead to an increase in ROS levels. Direct or indirect stimulation of ROS occurrence in the cell can lead to protein, nucleic acid, and membrane injury [46,47].

The antimicrobial spectrum of gold nanoparticles can be attributed to two features, the release of metal ions from the nanoparticle and the peculiarity of the nanoparticle per se. Simultaneously, the type of metal likewise, the characteristics of the particular nanoparticle cooperates with the observed antimicrobial spectrum. For instance, the smaller nanoparticles with a higher surface area to volume ratio are commonly observed to be more hazardous to bacteria, very likely due to their ability to pass through the cell membrane. The rate of metal attenuation also plays a great role in the degree of toxicity as nanoparticles with a faster rate of dissolution are often interdependent with increased noxious effects. Broadly speaking, metal ions appear to be the main source of toxicity to the microbe with nanoparticles characteristically acting as dexterous vehicles that agglomerate metal ion release at the cell surface [48,49,50].

Gold salt (tetrachloroaurate (III) trihydrate) as a synthetic metal includes a very efficacious antimicrobial spectrum to the heterogeneity of pathogens growing in the wound milieu, Figure 1. Gold NPs (AuNP) have been investigated as being notable in wound renovation as they have limited cytotoxicity, although their high cost would restrict their widespread utilization. There are a few noteworthy instances of their exploitation in wound healing. AuNPs along with a hydrogel of epigallocatechin gallate and α-lipoic acid are proven to be highly anti-inflammatory and potent antioxidants in wound healing. AuNPs in agreement with collagen demonstrates skin wound healing properties in a dose-dependent manner. Many studies reveal that the demonstration of the hydrocolloid membrane coated with gold-nanoparticles significantly mitigates the wound healing process. Different investigations concerning characteristics of AuNP showed the anti-oxidative and anti-microbial features of AuNPs, proving a very potent aspect in the regeneration of damaged collagen fibers and augmentation of wound healing course. AuNPs hasten the properties of the wound healing process through anti-inflammatory and anti-angiogenic activity by enhancing the emission of vascular endothelial growth factor (VEGF) IL-12, IL-8, and TNF-α [51,52,53,54].

## 6. Conclusion and Future Aspect

Resistance of bacteria against antibiotic treatment has been a troublesome global aspect of the medical industry, which comprises wound care and different issues related to the nature of wounds. Non-healing wounds after extensive operations remain still a serious challenge for the operation team, and a clinical investigation reveals that the process of healing has a direct correlation with wound chronicity. Thus, the cost of the complex management of wound treatment, related to biofilm, novel aspects as dispensable treatment strategies for the successful anti-biofilm course are indispensable for further approaches such as nanotechnology, and the development of nanoparticles with unquotable features. Recent technology empowers the creation of wound dressings for the use of their active molecules or remedies to the wound milieu. Quality of data have been researched purposefully to ensure the availability of evidence about the prosperous effect of silver nanoparticles compatibility and nano-structural materials and devices [55].

## Figures and Tables

**Figure 1 biomedicines-10-00679-f001:**
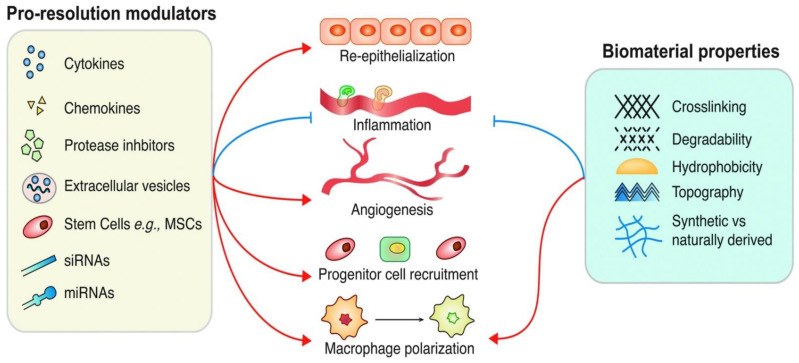
Therapeutic strategies target: Red-acceleration, Blue-Inhibition. Reproduced from Wilgus TA. Immune cells in the healing skin wound: influential players at each stage of repair. Pharmacol. Res. 2008; 58:112–116. and Tan J, Wu J. Current progress in understanding the molecular pathogenesis of burn scar contracture. Burns Trauma 2017; 5:14.

**Table 1 biomedicines-10-00679-t001:** Biomaterial-based immunomodulators, reproduced from Wright JB, Lam K, Buret AG, Olson ME, Burrell RE. Early healing events in a porcine model of contaminated wounds: effects of nanocrystalline silver on matrix metalloproteinases, cell apoptosis, and healing. Wound Repair Regen 2002; 10:141–151.) and Forlee M, Rossington A, Searle R. A prospective, open, multicenter study to evaluate a new gelling fiber dressing containing silver in the management of venous leg ulcers. Int Wound J 2014; 11:438–445.

Pharmaceutical	Effect	Outcomes	Species
Silver and Gold dressings	Antimicrobial, Inhibit MMPs, Formation of granulation tissue	Augumentation of wound healing	Daily Practice available
Si-RNA	Regulation of immune cell infiltration + Subside protease activity	Amelioration of wound closure	Diabetic mouse
MiR-99	Regulation of PI3k/AKT pathway	Increase wound closure	Diabetic mouse
MALP-2	Infiltration of macrophages and activation	Influence early wound closure	Diabetic mouse
PRP gel	Source of cytokines and growth factors	Boost healing in diabetic ulcer	Phase I

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
