# Peer review of "Benefit of Silver and Gold Nanoparticles in Wound Healing Process after Endometrial Cancer Protocol"

_biomedicines, 2022, doi:10.3390/biomedicines10030679_

Round 1

Reviewer 1 Report

The authors have made an interesting article describing utilization of silver and other synthetic molecules in wound healing process after endometrial cancer protocol, but some improvements should be made in order to be published in Biomedicines:

  1. In the Abstract I consider that the aim of this review is not clearly depicted and also it does not have a conclusion or a perspective, the abstract ends abruptly. Please rephrase the last phrase from the introduction.
  2. All the references are not written according to the journal’s guidelines. The point should be after the indication of the number, not before. Please revise all document.
  3. In the introduction
  • Ag +, the + should be superscript;
  • In the expression“nanoparticle silver ( AgNP)”, the space after the bracket should be deleted;
  • “Petrie agar” should be replaced with Petri;
  • aures, P.aureginosa, and E.coli should be written in italics, also all the species should be written in italics…revise all document.
  • AgNO3 should be written AgNO3, with 3 subscript.

4. In the 2nd part. Wound Healing process after operation.

  • Some paragraphs about the phases of wound healing lack from the review, please cite https://www.mdpi.com/1999-4923/12/10/983.
  • ( endometrial cancer protocol), ( removal of uterus…), ( diabetes, vascular disease, neuropathy, ….) should be written without space after the parenthesis.

5. It would be better that the silver nanoparticles and gold nanoparticles should be represented in a comparative table in order to make differences from different references and to emphasize the role of each formulation.

6. Please insert at least one Figure because it is hard to read.

Author Response

In my systemic review, I would like to gather the most crucial aspect concerning the novel use of silver and gold nanoparticles utilized in the wound endometrial treatment protocol.

There are still not enough publications associated with this type of management, that's why the references are really tough to find. 

The name of bacteria ( During Medical Study) all professors require Latin not Italic names of bacteria. 

 In the 2nd part. The wound Healing process after operation.

The issue related with 

  • Some paragraphs about the phases of wound healing lack from the review, please cite https://www.mdpi.com/1999-4923/12/10/983.
  • ( endometrial cancer protocol), ( removal of uterus…), ( diabetes, vascular disease, neuropathy, ….), I can't cite this information from the articles, though it is from my own observations during medical oncologic practice.

Reviewer 2 Report

The manuscript by Toczek et al. is intended as a systemic review about the possibility to use silver synthetic molecules (e.g. silver) in wound healing process after endometrial cancer protocol. The topic of the review is worthy of investigation and well fits within the aim and scope of “biomedicine” journal.

Nevertheless, publication is premature at this stage. Authors are encouraged to modify the paper by taking into consideration the following suggestions:

The main lack of the paper is the absence of a summarizing table reporting the main outcomes of the discussed literature works. This would greatly help readers in understanding the advantages and/or limitations of each strategy/materials. 

In the title, authors claimed about “other synthetic molecule”, while main focus on Silver and Gold-based materials. Of course this reviewer agrees with them that these are the most important classes of molecules, but authors should consider the possibility to modify the title for consistency.

In paragraph 5 (Gold nanoparticles use as antimicrobial synthetic agent), authors mentioned “Some metal ions” (page 5) and “certain nanoparticles” (page 6). They should explain which nanoparticle and modify the title paragraph for consistency.

Author Response

In my systemic review concerning silver and gold, I would like to explain mainly the mechanism of action related to gold and silver nanoparticles which is still pretty novel and not fully utilized therapy in the wound recovery course. That is why I was trying to depict the most crucial points from certain paper materials.  I performed mentioned alterations in the text.

Reviewer 3 Report

Title

Change tittle ‘Utilization of silver and synthetic molecules in wound healing post endometrial cancer treatment’, this is personal opinion.

Abstract

Opening sentence is too long and awkwardly phrased, please rewrite it something like ‘Silver and its representants, such as nanoparticles, and other resemble ingredients, are capable of considerably improving wound healing after extensive procedures, so they should be incorporated into current gynecology practices’. I suggest considering rewriting whole abstract as many sentence are difficult to understand, some examples: infectious offending agents to occur; silver as our allies; infectious offending agents to occur etc.

Introduction

Overall English is major concern and this manuscript cannot be considered without improving writing. As mentioned in previous comments, not only abstract but introduction is also full of words, which ether make no sense or awkwardly phrase. For example: The Serviceableness of silver.

Cite a latest article https://doi.org/10.1016/j.msec.2021.112592 to support the statement ‘There are cytotoxic to many bacterias, viruses, fungi, and even yeast which can disturb the healing process.’

Change bacterias to bacteria in above sentence.

In a review article, Scientific tables and graphs can be utilized to represent sizeable numerical or statistical data in a time- and space-effective manner. Readers are often drawn towards tables and figures, because they perceive it as easy-reading, as compared to reading a verbose account of the same content. This is totally missing in this review, I suggest to add:

Add graphic regarding biomaterial and molecular-based strategies to resolve general wounds principally focus on promoting reepithelialization, angiogenesis, progenitor cell recruitment, directing macrophage polarization, and inhibiting the migration of excess inflammatory cells. Current strategies are very diverse, either directly delivering cytokines and growth factors to the wound, or by using siRNAs, miRNAs, stem cells, and EVs to alter cytokine expression and production by cells in the wound bed. Material-based strategies are also being explored, particularly for their potential to direct macrophage polarization. Blue arrows indicate inhibition and red arrows indicate induction. EV, extracellular vesicle; siRNA, small interfering RNA.

Add a table showing the Biomaterial-based immunomodulatory strategies to reduce scarring. Add rows/column with compound, effect (such as wound closure vs. microbicidal), animal model type etc. listing potential immunomodulatory and silver based therapeutics for chronic wounds healing in post endometrial cancer.

Method

Add how literature search was conducted, which keywords were used, what inclusion exclusion criteria was adopted for deciding the literature to be included in review process.

Remove section 5. Gold nanoparticles use as antimicrobial synthetic agent.

Generally, the conclusion or closing remark of a review must provide a strong take home message to readers and I found that section is written with general overview. I suggest authors to include latest evidence of machine learning and artificial intelligence (AI) boosting the in silico methods in nanomedicine delivery citing latest development on the topic as reported in https://doi.org/10.1021/acssuschemeng.1c02589. The report demonstrates the latest development in nanomedicine, which will raise the outlook impact of the article.

Author Response

In my systemic review concerning the wound healing course, I would like to mention the newest issue related to this novel concern. As concerning the English language I would like to use, an outstanding word, though it ain't,  I will constantly re-use the same words, and like we know it's not desirable

Mentioned graphs associated with biomaterial and molecular-based strategies it is really hard to search and rely on either.

Round 2

Reviewer 2 Report

The paper can be now published in its current form

Author Response

Concerning my systemic review, I alter a few aspects associated with adding some graphs and one table related to the immunomodulators biomaterial in the wound healing course). Lastly, I was focusing on the minor English revision.

Reviewer 3 Report

Author considered few of comments but manuscript is still in poor shape.

We suggested changing title as it is ambiguous. `Utilization` might confuse readers that authors are showing utility of gold & silver in this paper in wound healing although it is a review paper

I suggested and even rephrased few sentence to follow but author did not considered. For example, Abstract opening sentence is covers 264 character spanning 3.5 line, which severely dilutes the understanding of the article.

I suggested to add graphics and tables, I see authors added 2 tables with silver and gold which seems directly copied from the journal article without taking permission from journal as no statement appears in table caption. While source for silver is cited as references 34-35 but no source is cited for the gold table. Both of these tables have obsolete information. I suggested few refs lately published in context with the content of the review to improve the paper (https://doi.org/10.1016/j.msec.2021.112592 & https://doi.org/10.1016/j.msec.2021.112592) but authors completely ignored. Just citing the refs without permission is the infringement of copyright.

Copying the tables and adding information from randomly does not make an impactful paper. Author did not include any graphic as suggested to improve the review giving an aspect of mechanistic role of silver. Remember, MDPI biomedicine is a high impact journal and without considering proper revision, paper shall not be considered.

Author Response

Concerning my systemic review, aforementioned, I alter a few aspects related to the title, as well as proceeded with the abstract change for a shorter version. As you recommended  I cite the article from the link as you instruct. Chiefly I focused on making a graph and table associated with immunomodulators and material-based strategies.

  1. I’ve altered the beginning of the title for more comfortable pronunciation.
  2. I performed citation of the certain literature which was recommeded by reviewer 3 (Kanchan M.Joshi,AmrutaShelar,UmeshKasabe, et al.: Biofilm inhibition in Candida albicans with biogenic hierarchical zinc-oxide nanoparticles ; Materials Science and Engineering: C; Available online 3 December 2021, 112592)
  3. In every single case of bacteria name, I utilized latin language.
  4. I did shorter abstract.
  5. Small meritoric changes throughout the text
  6. Table 1- Comparision of the biomaterials.
  7. figure 1 deliniated , inhibition or induction of inflamation.
  8. Point 6,7 was recommended from the reviever 3 to augument article by extra-novel information.

Round 3

Reviewer 3 Report

accept in present form